# Identification of Novel Mutations in Colorectal Cancer Patients Using AmpliSeq Comprehensive Cancer Panel

**DOI:** 10.3390/jpm11060535

**Published:** 2021-06-09

**Authors:** Bader Almuzzaini, Jahad Alghamdi, Alhanouf Alomani, Saleh AlGhamdi, Abdullah A. Alsharm, Saeed Alshieban, Ahood Sayed, Abdulmohsen G. Alhejaily, Feda S. Aljaser, Manal Abudawood, Faisal Almajed, Abdulhadi Samman, Mohammed A. Al Balwi, Mohammad Azhar Aziz

**Affiliations:** 1King Abdullah International Medical Research Center, Medical Genomics Research Department, Ministry of National Guard Health Affairs, King Saud Bin Abdulaziz University for Health Sciences, Riyadh 11481, Saudi Arabia; BalwiM@NGHA.MED.SA; 2King Abdullah International Medical Research Center, Saudi Biobank, King Saud Bin Abdulaziz University for Health Sciences, Ministry of National Guard Health Affairs, Riyadh 11481, Saudi Arabia; alghamdija@ngha.med.sa (J.A.); SayedA@NGHA.MED.SA (A.S.); 3Department of Pathology, College of Medicine, Imam Mohammad Ibn Saud Islamic University (IMSIU), Riyadh 13318, Saudi Arabia; Alhanoufomani@hotmail.com; 4Clinical Research Department, Research Center, King Fahad Medical City, Riyadh 11564, Saudi Arabia; saghamdi@kfmc.med.sa; 5Comprehensive Cancer Center, King Fahad Medical City, Riyadh 11564, Saudi Arabia; aalsharm@kfmc.med.sa; 6King Abdul Aziz Medical City-National Guard Health Affairs (NGHA), King Abdullah International Medical Research Center, King Saud Bin Abdul Aziz University for Health Sciences (KSAU-HS), Riyadh 14611, Saudi Arabia; shieban@hotmail.com; 7Faculty of Medicine, King Fahad Medical City, Riyadh 11564, Saudi Arabia; aalhejaily@kfmc.med.sa; 8Department of Clinical Laboratory Sciences, Chair of Medical and Molecular Genetics Research, College of Applied Medical Sciences, King Saud University Riyadh, Riyadh 11564, Saudi Arabia; faljaser@ksu.edu.sa (F.S.A.); mabudawood@ksu.edu.sa (M.A.); 9Department of Clinical Laboratory Sciences, College of Applied Medical Sciences, King Saud Bin Abdulaziz University for Health Sciences, Ministry of National Guard Health Affairs, Riyadh 11481, Saudi Arabia; Faisal.almajed@gmail.com; 10Department of Pathology, Faculty of Medicine, University of Jeddah, Jeddah 23218, Saudi Arabia; dr.a.samman@hotmail.com; 11King Abdullah International Medical Research Center, Colorectal Cancer Research Program, Department of Cellular Therapy and Cancer Research, Ministry of National Guard Health Affairs, King Saud Bin Abdulaziz University for Health Sciences, Riyadh 11481, Saudi Arabia

**Keywords:** colorectal cancer, personalized medicine, biomarker, AmpliSeq

## Abstract

Biomarker discovery would be an important tool in advancing and utilizing the concept of precision and personalized medicine in the clinic. Discovery of novel variants in local population provides confident targets for developing biomarkers for personalized medicine. We identified the need to generate high-quality sequencing data from local colorectal cancer patients and understand the pattern of occurrence of variants. In this report, we used archived samples from Saudi Arabia and used the AmpliSeq comprehensive cancer panel to identify novel somatic variants. We report a comprehensive analysis of next-generation sequencing results with a coverage of >300X. We identified 466 novel variants which were previously unreported in COSMIC and ICGC databases. We analyzed the genes associated with these variants in terms of their frequency of occurrence, probable pathogenicity, and clinicopathological features. Among pathogenic somatic variants, 174 were identified for the first time in the large intestine. APC, RET, and EGFR genes were most frequently mutated. A higher number of variants were identified in the left colon. Occurrence of variants in ERBB2 was significantly correlated with those of EGFR and ATR genes. Network analyses of the identified genes provide functional perspective of the identified genes and suggest affected pathways and probable biomarker candidates. This report lays the ground work for biomarker discovery and identification of driver gene mutations in local population.

## 1. Introduction

Colorectal cancer (CRC) is a heterogeneous disease. Inter-patient heterogeneity has been one of the major obstacles towards developing therapeutic strategies. Different populations have been found to show varied response towards standard of care regimens [1]. This variation has largely been attributed to the difference in underlying gene mutations and genetic changes which determines the progression of CRC. CRC progresses with continuing accumulation of genomic and epi-genomic alterations, which eventually induce oncogenic transformation of the normal colon cell into tumor cells followed by metastasis. Pathways responsible to initiate CRC are well known, based on the evidence of mutations and chromosomal changes observed in patients. The mechanistic role of signaling pathways in causing CRC has constantly been enriched with better understanding of the underlying gene mutations. These gene mutations have been used as biomarkers to predict disease progression and outcome of therapeutic regimens.

KRAS mutation status is routinely used for administering antibodies to inhibit epidermal growth factor receptor (EGFR). Successful use of these antibodies (cetuximab and panitumumab), only in KRAS wild-type patients, had set the stage of precision and personalized medicine. However, not all patients with the wild-type KRAS gene respond to anti-EGFR therapy. Therefore, there is a pertinent need to identify biomarkers that can capture the population heterogeneity and facilitate the practice of precision and personalized medicine. Earlier studies have taken up population-based mutational profiling of CRC to develop the concept of precision medicine [2,3]. Population-specific mutational analysis of colorectal cancer is scarce in Saudi Arabia but highly pertinent to develop the precision and personalized medicine paradigm [4,5,6]. With the technological advancement in detecting mutations at an unprecedented scale, the possibility of practicing precision medicine through biomarkers has further increased. Precision medicine in colorectal cancer is more relevant than in other cancers, owing to its heterogeneity in development as well as response to therapy. Colorectal cancer metastasis has been largely unimproved with new developments in therapeutics, and precision medicine holds optimism especially for these patients [7,8]. There is better precision and accuracy in detecting mutations in patients that can be used as predictive and prognostic biomarkers. Use of non-invasive biomarkers for colorectal cancers is one of the most promising strategies in treating CRC [9].

Next-generation sequencing (NGS) technology can be used with DNA-enrichment methods to generate deep sequencing of target genes or genomic regions of interest, such as the exome or identified cancer hotspots. For the targeted detection of mutations in known cancer genes, a comprehensive cancer panel (IonAmpliSeq) is available. Gene panels allow simultaneous detection of relevant mutations with unprecedented accuracy and sensitivity. This comprehensive cancer panel (CCP) is designed to target coding DNA sequences (CDS) and splice variants from 409 tumor suppressor and onco-genes that are frequently mutated. The requirement of small amount of input DNA (only 40 ng) per reaction enables challenging analysis of formalin-fixed, paraffin-embedded (FFPE) tissues. The use of the IonAmpliSeq™ Cancer Panel and NGS using the IonTorrent platform provides a fast, easy, and cost-effective sequencing workflow for detecting genomic hotspot regions that are frequently mutated in human cancer. A previous study from Jeddah, Saudi Arabia used the Ion AmpliSeq™ Cancer Hotspot Panel v2 which spans only 50 frequently mutated genes [10].

In this study, we used IonAmpliSeq™ CCP to sequence samples from 99 archived patient samples from two hospitals in Riyadh, Saudi Arabia, over a duration of two years. The confirmation of well-known mutations point towards the chromosomal instability pathway as the predominant mechanism of the development of CRC in this cohort. We provide comprehensive analyses of novel variants that can be useful for biomarker discovery and identification of driver genes. Discovery of biomarkers and identification of driver genes from local population is critical in developing precision and personalized medicine approach towards addressing colorectal cancer.

## 2. Materials and Methods

### 2.1. Patient Description and Sample Collection

A total of 100 patient tumor samples were retrospectively recruited in this study, and, after exclusion of 1 sample due to low DNA quality, we sequenced 99 samples, and clinicopathological characteristics were available from 95 patients. Sequencing data from 90 of these samples qualified for coverage requirement and was used for further analyses. All samples were collected in the period between 2016 and 2018 at King Abdulaziz Medical City (KAMC) and King Fahad Medical City (KFMC), Riyadh, Saudi Arabia. All samples were diagnosed as primary colorectal adenocarcinoma at histopathology level. Patients were excluded if: (i) they had been treated with chemotherapy or radiotherapy prior to tumor resection, (ii) they had familial adenomatous polyposis (FAP) or hereditary non-polyposis colorectal cancer (HNPCC), or (iii) the formalin-fixed, paraffin-embedded tissue (FFPE) samples, patients’ clinical and pathological data, or written informed consent form signed by patient to access the archival samples were not available.

The formalin-fixed, paraffin-embedded tissues (FFPE) blocks from patients with colorectal adenocarcinomas were retrieved from the archives of the Department of Anatomical Pathology Laboratory in KAMC and KFMC. All slides were revised and marked by a histopathologist before DNA extraction. We selected only marked tissue with tumor percentage more than 40% and used 1–2 slides for extraction based on tissue size. Chart reviews were done after obtaining the ethical approval to collect the demographic and clinicopathological features from the hospital information system BESTCare 2.0 A at KAMC, including age at diagnosis, gender, tumor stage, site, and metastasis grade.

### 2.2. Ethical Approval

Full Institutional Review Board (IRB) approval was given by King Abdullah International Medical Research center (KAIMRC), Ministry of National Guard, Health Affairs (IRB protocol #RC13/249/R). All patients’ data were secured and accessed only by research investigators.

### 2.3. DNA Extraction

Genomic DNA was extracted from FFPE samples that were assessed by a pathologist to select the appropriate block to assure presence of colorectal cancer cells and excluded the insufficient necrotic tissue for NGS. DNA was extracted either from slide sample using Ion AmpliSeq ™ Direct FFPE DNA Kit (Thermo Fisher Scientific Inc, Toronto, ON, Canada) according to the manufacturer’s instructions. In case of FFPE block samples, DNA was extracted from FFPE blocks using 8µm of tissue ribbon using QIAamp DNA FFPE Tissue Kit (QIAGEN) following manufacturer’s instruction. Measurement of the DNA quality and concentration was done by using Qubit^®^ 3.0 Fluorometer (Life Technologies).

### 2.4. Comprehensive Cancer Panel (CCP) and Data Availability

The pre-designed comprehensive cancer panel (CCP) from Ion AmpliSeq™ (Life Technologies) was used. This panel comprises 16,000 primer pairs in four primer pools for 409 genes, which covers approximately 15,749 somatic mutations reported in The Catalogue of Somatic Mutations in Cancer (COSMIC). For the complete list of 409 genes, see Appendix A. All sequencing data generated from 90 patients is deposited in SRA database (reference PRJNA685957, https://www.ncbi.nlm.nih.gov/sra/PRJNA685957 Last accessed on 1 June 2021)

### 2.5. Library Preparation and NGS Data Analysis

The library was constructed using Ion AmpliSeq™ (CCP) Library Kit 2.0 (Life Technologies) and Ion Xpress™ Barcode Adapter 1–16 Kit (Life Technologies) according to manufacturer’s instructions. Library quantification was done using the Ion Library TaqMan Quantitation Kit (Life Technologies) following standard procedure available. The qualified library was sequenced by the use of Ion S5XL Semiconductor Sequencer following the manufacturer’s user guide.

### 2.6. Variant Calling and Annotation

Variants were called by Torrent Suite Variant (TSV) (version 5.8) [11]. Variants with a coverage of more than 300X and read quality more than 50 were included in this study to enhance the quality of identified somatic variants. Variants that passed this quality metrics were annotated by using Ensemble Variant Effect Predictor (VEP) tool (version 102). This tool uses gnomAD (version r2.1) and the Catalogue Of Somatic Mutations (COSMIC) databases (version 90) [12]. We excluded common variants previously reported in Ensemble (v102) and only included variants classified as confirmed somatic or pathogenic by COSMIC database. This classification is based on functional analysis through hidden Markov models (FATHMM). Further, variants were classified into colorectal cancer associated or other organ sites. The potential damaging effect at protein level of the variants were assessed using prediction software using Sorting Intolerant From Tolerant (SIFT; v5.2.2) and Polymorphism Phenotyping v2 (PolyPhen2; v2.2.2) scores [13,14]. These scores predict the impact of detected missense variants on the human protein structure. All variants which showed as deleterious on SIFT and/or damaging on PolyPhen2 were included for downstream analysis.

### 2.7. Molecular Profiling and Statistical Analysis

Descriptive statistics were applied to summarize patient characteristics based on clinicopathological features. Summary statistics of the identified genetic variants were carried out in PLINK [15] to calculate the minor allele frequencies (MAF) and Hardy–Weinberg equilibrium *p*-value [3]. Associations between mutations and CRC or histological features were determined using Fisher’s exact test. Due to the limited sample size, tumor stages were grouped into early (stage I–II) and late (stage III–IV). Site of tumor was classified as left, right, and others. The involvement of lymph nodes and secondary metastases were analyzed as a dichotomous trait. All analyses were conducted using JMP Prostatistical software (JMP^®^, Version 13. SAS Institute Inc., Cary, NC, 1989–2019). Sequence Kernel Association—Optimal unified test (SKAT-O) was used to perform gene-based association analysis [16]. The association of rare variants with tumor stage (defined as late versus early), gender (female versus male), age group (young < 50 years versus old), and tumor location (left versus right) was analyzed. The variants were weighted based on their allele frequency, where rare variants were assigned higher weight than common variants. To account for multiple testing, an adjusted *p*-value of 0.0001 was considered as a significant threshold, reflecting the Bonferroni correction of 409 genes.

### 2.8. Ingenuity Pathway Analysis

The networks for mutated genes were generated through the use of IPA (QIAGEN Inc., https://www.qiagenbio-informatics.com/products/ingenuity-pathway-analysis. Last accessed on 1 June 2021) [17]. Networks were created using following filter: Species = Human AND Disease = Cancer AND mutation = hemizygous OR in-frame OR gain-of-function OR frameshift OR missense OR homozygous OR null mutation OR silent OR heterozygous OR loss of function OR knockout OR nonsense. Two networks were generated—one with the 27 most frequently mutated genes and another with 75 genes harboring pathogenic mutations reported in large intestine. The connect function was used to investigate the known interactions among these genes. The overlay function was used to find the association of these genes with canonical pathways and finding candidate biomarkers.

## 3. Results

### 3.1. Cohort Characteristics

The baseline characteristics of the analyzed samples are shown in Table 1. The median age of patients was 62 years, with 58 of them being male (61%). According to TNM staging system, 65% of the patients were classified as T3, with 59% showing no spread to regional lymph node (T0), and 96% were without distant metastases. The highest proportion of patients were diagnosed as stage III (39%), and more tumors were located in left colon (52%).

### 3.2. Novel Variants Identified in Colorectal Cancer Patient Cohort

From a panel of 409 genes, we identified 4256 variants. Among these, 483 variants were classified as novel, as they were not found in the COSMIC database. However, 17 of these variants were reported in the international cancer genome consortium (ICGC) database. All novel variants are provided in Appendix A. We checked for the probability of these variants to be germline by analyzing their variant allele frequency (VAF). A total of 69 variants presented in at least one patient with a VAF between 49–51 or 99–100, indicating that they could be germline mutations, which is also supported by the MAF (>1%) among 45 of these variants (Appendix A).

Among the 4256 identified variants, 299 variants were classified as pathogenic. A total of 174 variants from 299 pathogenic variants were found to be identified for the first time in the large intestine, representing novel variants in colorectal cancer (Figure 1 and Appendix A). We employed two different methods (SIFT and PolyPhen) for classifying 561 somatic variants. Both methods suggest the detected variants to be either synonymous (*n* = 240) or missense (*n* = 247) (Figure 2A). According to the PolyPhen scoring method, 143 mutations were predicted to be benign, and the rest could be pathogenic (Figure 2B). The SIFT prediction method also provided similar categorization, with 111 variants listed as tolerated, and 130 variants were classified as deleterious (Figure 2C).

### 3.3. Novel Variants Identified in Most Commonly Mutated APC, RET, and EGFR Genes

The highest mutated genes (*n* = 20) among the patients were identified based on the presence of at least one confirmed pathogenic variant and arranged according to decreasing trend of frequency in the patient cohort (Table 2). A total of 96% of the patient samples had at least one confirmed pathogenic variant within the APC gene. We identified 5 novel (defined as previously unreported in COSMIC database) variants out of total 38 variants detected in the APC gene. These novel variants include c.1696G>A (p.V566I) missense mutation at exon 14, c.1697delT (p.V566X) frame shift mutation at exon 14, c.2680_2681delGTinsTA (p.Val894Ter) stop gain mutation at exon 16, c.3917delA (p.E1306X), frame shift mutation at exon 16, and c.4320-4341del ACCACCTCCTCAAACAGCTCAA (p. PPPPQTAQ1440-1447X). A total of 23 of 38 variants were confirmed as somatic variants in COSMIC. Of the 23 variants, 15 were confirmed as pathogenic, and 12 were confirmed as tissue-specific pathogenic variants for the large intestine. RET gene mutations were found in 53% of the patient samples. Out of 13 detected variants in RET gene, 3 were somatic and one of was pathogenic (p.L769 = 0.53% patient samples harbored EGFR gene mutation. A total of 17 variants were detected, of which 6 were somatic, and 2 variants were specific for the large intestine. One of these variants was a high-impact nonsense mutation (p.R1068*). This comprehensive analysis and finding of novel variants within known genes could open up avenues to develop biomarkers that will be relevant for local population.

### 3.4. Colorectal-Cancer-Specific Variants Mapped to Twenty-Seven Genes

The distribution of pathogenic mutations found in the large intestine across gender, age, tumor stage, site, lymph node, and metastasis is described in Figure 3. A total of 73 variants specific for colon and rectum were identified within 27 genes. Tissue-specific pathogenic variants in the studied population show that the APC gene was the highest mutated, with variants detected in 66% of the samples, followed by ERBB2 (51%), ATR (45%), EGFR (40%), and FGFR3 (30%) genes. It is known that APC gene mutation is the initial event in CRC progression and is well depicted in our results. We observed variants in APC, ATR, KRAS, ATM, and KIT genes in the left colon of young female patients (<50 years age) in stage 1. However, no mutation was observed in young male patients in the left colon in early stage (I and II), but mutations were observed in these patients in the right colon and rectum. This detailed catalogue of variants analyzed according to clinicopathological features could be further used for molecular classification of patients.

### 3.5. Left Colon Exhibits Higher Mutation Load

We identified 27 genes with at least one confirmed pathogenic variant presented in at least three patients. We found that patients with left side of the colon had higher prevalence of mutated genes, with the exception of ATR, MLH1, ATM, MTOR, PDGFRA, EP300, COL1A1, PTEN, and TSHR genes (Figure 4A). A significantly higher number of mutations were observed in FGFR3 gene in the left side and EP300 and TSHR genes on the right side of the colon. While comparing the early- and late-stage tumors, the prevalence of mutated genes were almost similar except for significantly higher COL1A1 gene mutations among patients in early stage when compared to late stage (Figure 4B).

### 3.6. Pathogenic Variants in ERBB2 Were Significantly Correlated with Mutations in EGFR and ATR

Gene correlation analysis showed that occurrence of pathogenic gene mutations was correlated (Figure 5). Presence of pathogenic variants at ERBB2 was significantly correlated with mutations in EGFR and ATR (r^2^ = 0.39 and 0.26; *p*-values = 0.0001 and 0.01, respectively). High correlation was found between KDM6A and UBR5 gene mutations (r^2^ = 0.47, *p*-value = 2.3 × 10^−6^). FGFR3 gene was the most correlated. It was found positively correlated with HNF1A and TP53, whereas EGFR and ATR were negatively correlated.

We tested the association of mutated genes with clinicopathological variables and found a significant association between ERBB2 mutation and tumor late stage (Fisher’s exact t-test; *p* value = 0.04). Mutations in EP300 and TSHR mutations were found to be significantly associated with right colon tumor (chi-square; *p* = 0.02 and 0.01; respectively). Increased FGFR3 mutations was observed in the left colon (chi-square; *p* = 0.01)

For the gene-based rare variants analysis (SKAT-O), no gene was associated with clinicopathological variable at the significant threshold. However, suggestive significance was found between PIK3CB and colorectal cancer on the left side (*p* = 0.0007), androgen receptor (AR) and female gender (*p* = 0.0002), TGM7 and young patient (*p* = 0.002), and EXT1 and late stage (Table 3).

### 3.7. Network Analysis of Mutated Genes

Using ingenuity pathway analysis, we created an information-based network of 27 highly mutated genes and found TP53 was the most connected node (Figure 6A). This network identified 16 druggable target genes. A network of 75 genes with pathogenic mutations in the large intestine also exhibited TP53 as a highly connected node. Thirty-three of these network genes were identified as target molecules (Figure 6B). Both networks identified the TSHR gene as a potential druggable target (Appendix A).

Genes with pathogenic mutations in the large intestine were found to be associated with hepatic fibrosis signaling, CRC metastasis, senescence, NF-ĸB, and regulation of epithelial-to-mesenchymal transition pathways. Genes associated with these pathways are shown in Figure 6C. Biomarker analysis of these 75 genes revealed 16 candidate molecules, some of which are already in clinical use (Figure 6D and Table 4). These biomarkers have potential use in determining diagnosis, prognosis, efficacy, and response to drugs.

## 4. Discussion

Tumorigenesis and progression of cancer is suggested to be driven and supported by gene mutations [18,19,20,21,22,23]. Somatic mutations that are observed in cancer cells help to understand the cause and severity of the disease. Colorectal cancer is well known to have specific gene mutations associated with particular stages of the disease. In the present study, we aimed to provide comprehensive analysis of gene variants as studied in a cohort of patients in Riyadh, Saudi Arabia. We employed NGS on the eAmpliSeq comprehensive cancer panel to unravel the information locked in FFPE samples. This study provides successful evidence to support the use of archived samples and sequencing technology to generate information that is relevant for the local population. While we aimed to understand the mutational profile in the local population, we found results that confirmed existing evidence supporting the initiation and progression of CRC. We also report novel variants in our population, which is suggestive of a unique genomic landscape of patients and supports the idea of precision and personalized medicine [5,6,24].

As determined by two separate prediction methods (PolyPhen [14] and SIFT [25]), most of the detected mutations were missense and synonymous. This is in conformity with a recent pan-cancer analysis [26] and opens up avenues to further study the effect of point mutations in CRC. These point mutations could be responsible for changes in gene expression and mRNA secondary structures. Similar studies from other populations have also reported the predominance of synonymous and missense mutations [2]. However, the challenge to separately identify driver mutations from passenger mutations with precision and accuracy is still an ongoing area of intense research [27,28,29].

We identified the APC gene as highly mutated in our cohort, with less-common mutation frequency for RET, EGFR, LRP1B, and ERBB2 genes. The APC gene mutation is well known to be one of the very early events in initiation of CRC. RET gene fusions have been associated with a subtype of CRC on the right side of the colon. EGFR was earlier identified as one of the highest mutated genes in a cohort of patients from Jeddah, Saudi Arabia, and confirms our results [10]. This study adds to the evidence of detected variants in a previous similar study from a different geographical location. More studies from different regions of the country are needed, as there is an observed disparity in incidence and mortality of CRC in other regions within Saudi Arabia [4]. Identification of novel mutations in APC, RET, and EGFR mutations may lead us to develop predictive and/or prognostic biomarkers for CRC. Mutations in these genes were earlier studied in detail for their use as biomarkers [30,31,32,33]. Most of the previous studies are associated with the common mutation of the APC gene except two studies on an Arab cohort that showed APC mutation frequency was the second highest (34%) after P53 gene. Another study from gulf region patients showed 27.3% mutation frequency and higher mutation of TP53 (52.5%).

We found more CRC cases are localized at the left site compared to the right side or rectum site. This is suggestive of the involvement of the CIN pathway and is evident from our results, which show APC, KRAS, and P53 as highly mutated genes. However, PIK3CA do not appear among the highly mutated genes, but three pathogenic mutations were identified among PIK3CA (p.R88Q, p.I102F, and p.PI04L). All these mutations are reported in the large intestine except (p.I102F mutation). This could be due to the population-specific nature of the mutations and suggest further study to understand the mechanism of CRC progression in these patients. Left-sided colorectal cancers have better prognosis and response to 5-Fluorouracil-based and targeted therapies [34]. Our results are therefore very significant in understanding and predicting the prognosis of local patients who primarily exhibited mutations suggesting left-sided CRC. KRAS mutations, EGFR/HER2 amplifications, and a high level of amphiregulin and epiregulin expression has been observed in left-sided CRC [35,36]. Treatment strategies widely differ according to the location of the tumor, and hence understanding the molecular differences in local population is pertinent.

Increasing incidence of CRC cases in early ages has caused the guidelines for screening to be revised [37]. Our observation regarding young patients suggests presence of mutations in the left colon of female patients in stage I, whereas young male patients did not show any mutations in left colon in stage I and II. This can be an important finding that can be studied further in larger cohorts to develop early diagnostic tests. Our catalog of reported variants have enriched the database for CRC and would be useful in building up larger studies for finding actionable targets and biomarkers. Information regarding these variants will need to be complemented with further levels of evidence to prove their role in CRC or identify them as drug targets. Multiomics approach is therefore recommended to be carried out on the same samples for further proof of evidence [38,39].

The gene correlations observed in our cohort and network analysis could provide clues for the possible mechanism of CRC development. These networks and correlation analyses should be done at gene-expression level to further understand the mechanistic details and effect of variants [40]. Network analyses confirms the probable effect of the detected variants through well-known pathways. We report mutations that can be associated with senescent pathway and point towards development of therapeutic strategies. Targeting senescent pathways has been suggested as an anticancer therapy and point towards their role in senescence and metastasis. Biomarker candidate molecules need to be further validated and tested for advancing into a clinical setting.

Though our study is limited with a smaller number of patient samples, it does exhibit the heterogeneous nature of CRC [41]. Another major limitation of our study is lack of matched normal samples to account for possible germline mutations. This is one of the bargains for utilizing the treasure of formalin-fixed samples. Using a matched normal sample is a requirement for accurately classifying somatic mutations and ruling out germline mutations. However, the availability of matched normal tissue is a limitation when using archived and fresh samples [42]. Computational methods have been developed that are arguably better than matched normal tissue [43]. Most of the studies have relied on increasingly rich databases to identify novel mutations in absence of matched normal samples [44]. In order to address this issue, we used public databases and also employed an earlier reported method where the VAF corresponding to 50 or 100% may indicate their probability to be germline mutation [45].

This study provides evidence that can be useful for developing biomarker-based precision medicine as well as allowing us to appreciate the heterogeneity in CRC and hence develop strategies accordingly.

## Figures and Tables

**Figure 1 jpm-11-00535-f001:**
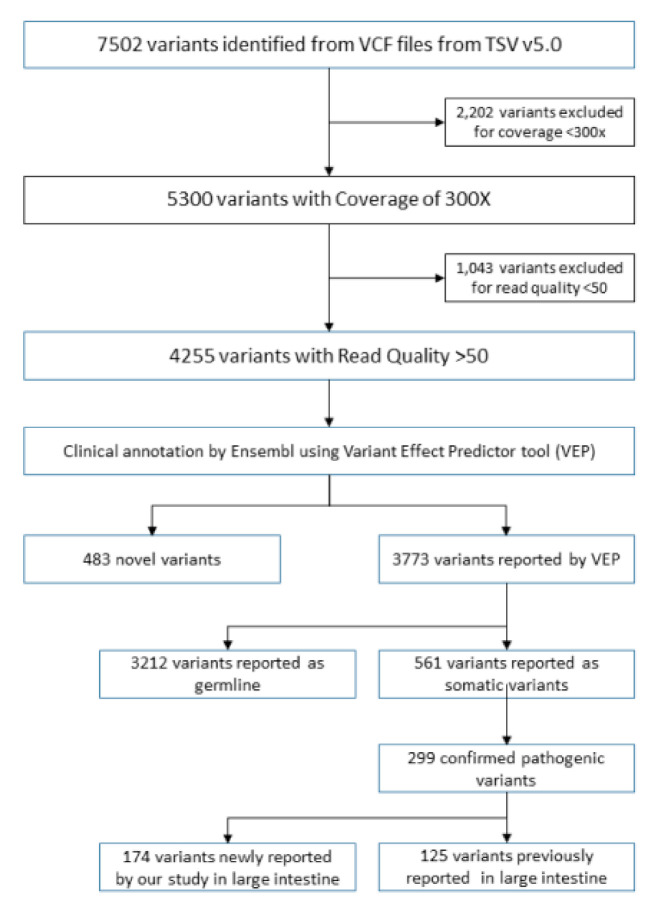
Variant filtration analysis workflow. Schematic illustration of variants identified in this study. A total of 483 novel variants were identified, and 561 somatic variants were observed. This study focused on pathogenic variants that were identified as novel in the large intestine.

**Figure 2 jpm-11-00535-f002:**
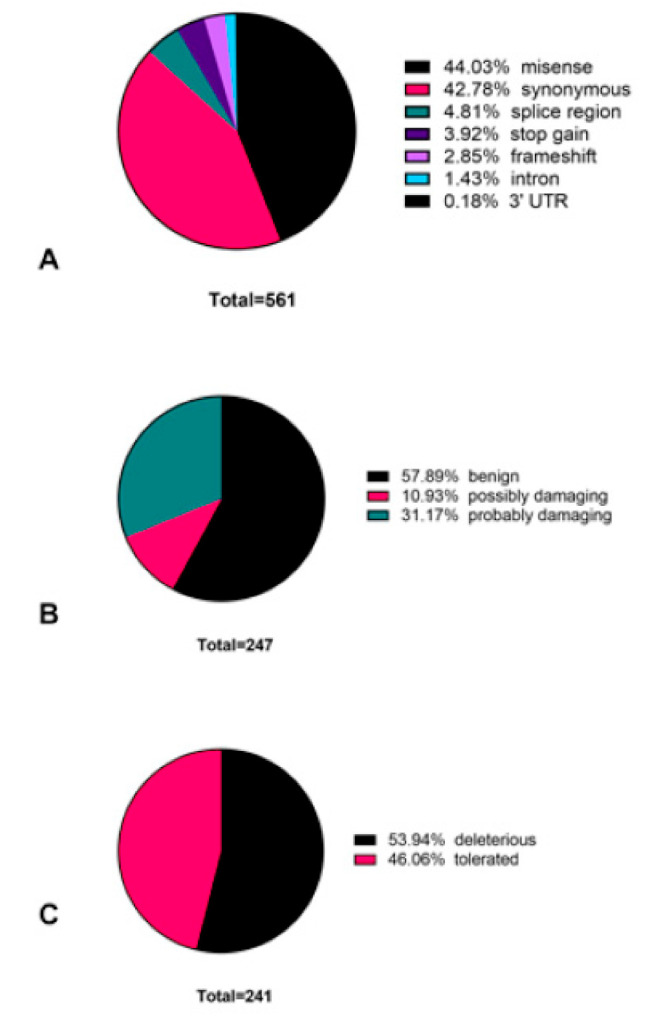
Classification of somatic variants. A total of 561 variants were classified by consequence (**A**), PolyPhen score (**B**), and SIFT score (**C**).

**Figure 3 jpm-11-00535-f003:**
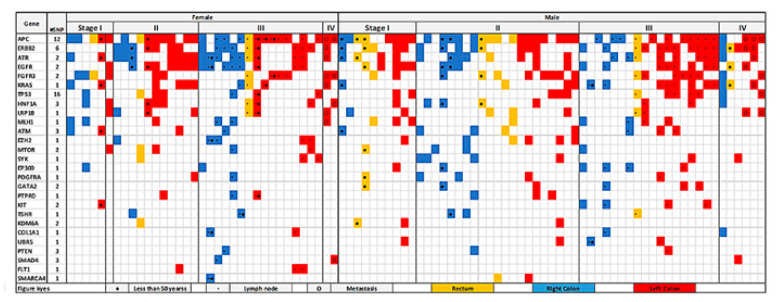
Mutation distribution based on gender, age, stage, site, lymph node, and metastasis. This figure shows only variants reported as pathogenic and located in the large intestine in COSMIC database and found in three individuals or more.

**Figure 4 jpm-11-00535-f004:**
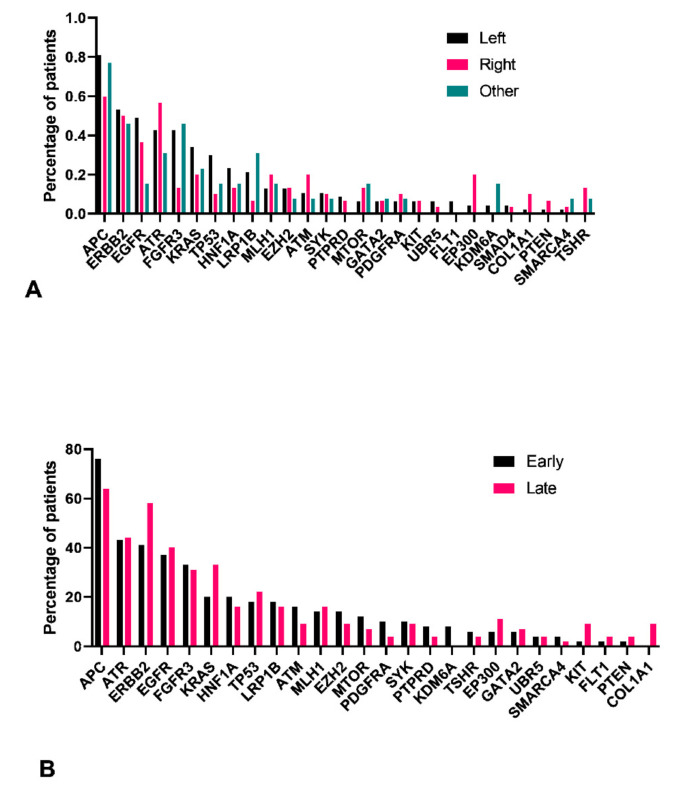
Frequency of variants in 27 genes among samples with at least one confirmed pathogenic variant for the large intestine. Frequency of variants based on tumor location (**A**) and stage (**B**). *Y*-axis denotes the number of samples with at least one confirmed pathogenic variant for the large intestine for that particular gene. For location, each bar is divided into left, right, and other categories, whereas for stage, they were grouped into early (stage I and II) and late (stage III and IV).

**Figure 5 jpm-11-00535-f005:**
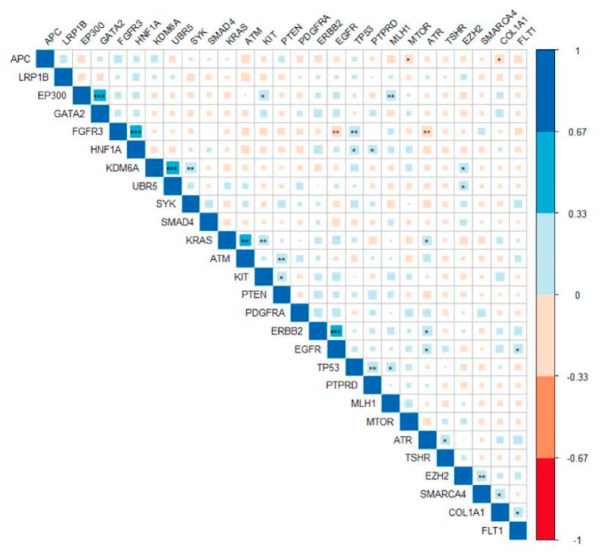
Correlation between mutated genes. The Pearson correlation between presence of a tissue-specific pathogenic variant between genes. Significant correlations are noted with * for *p*-values < 0.05, ** for *p*-values < 0.01, and *** for *p*-values less than 0.001. Color and size of the square denotes the value of correlation as indicated in the bar legend.

**Figure 6 jpm-11-00535-f006:**
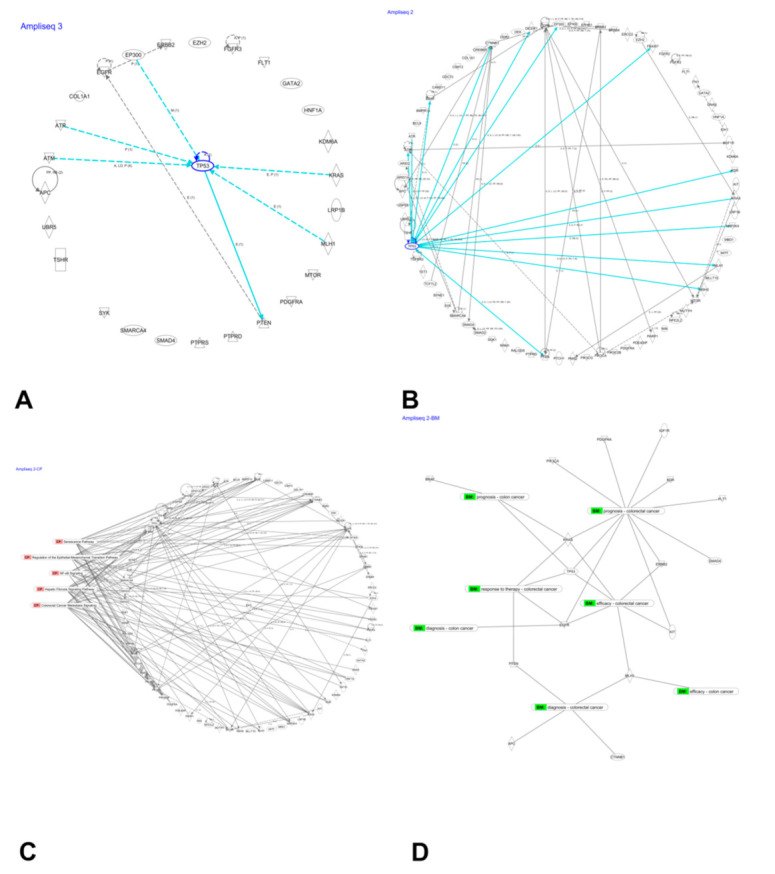
Network analyses of genes with reported variants. Network of 27 most frequently mutated genes (**A**). Network of 75 genes harboring pathogenic mutations reported in the large intestine (**B**). Association of 75 genes with canonical signaling pathways (**C**). Possible biomarker candidates for diagnosis, prognosis, efficacy, and response to drugs for colon and colorectal cancer (**D**).

**Table 1 jpm-11-00535-t001:** Clinicopathological features of CRC patients.

Age, Years (SD)	62 (14)
Male, n (%)	58 (61%)
Stage	
I, n (%)	17 (18%)
II, n (%)	32 (34%)
III, n (%)	37 (39%)
IV, n (%)	8 (9%)
Primary Tumor	
T1, n (%)	2 (2%)
T2, n (%)	18 (19%)
T3, n (%)	61 (65%)
T4, n (%)	13 (14%)
Lymph Node	
N0, n (%)	55 (59%)
N1, n (%)	33 (35%)
N2, n (%)	6 (6%)
Distant Metastasis	
M0, n (%)	90 (96%)
M1, n (%)	4 (4%)
Site	
Left colon, n (%)	47 (52%)
Right colon, n (%)	30 (33%)
Rectum, n (%)	13 (14%)

T, tumor; N, node (0, no nodes; 1, 1 node; 2, 2 nodes); M, metastasis (0, no metastasis; 1, metastasis); SD, standard deviation.

**Table 2 jpm-11-00535-t002:** List of twenty genes with variants in order of frequency in the sample cohort.

Description of Variants Variants and Individuals for the Top 20 Genes										
Gene	Variants								Individual ^a^	
	Total	Novel	Pathogenic		Somatic	PolyPhen Damaging	SIFT Deleterious	Non-Synonymous	Pathogenic %	Somatic %
			All Tissue	Tissue-Specific						
*APC*	38	5	15	12	23	1	2	30	0.96	0.99
*RET*	13	0	2	0	3	3	3	6	0.53	0.83
*EGFR*	17	0	4	2	6	0	0	7	0.53	0.69
*LRP1B*	70	12	10	1	22	3	1	48	0.52	0.86
*ERBB2*	14	3	3	3	4	1	2	10	0.51	0.52
*ATR*	31	6	4	2	11	1	0	19	0.46	0.68
*CSMD3*	42	5	4	0	8	5	3	34	0.44	0.52
*RALGDS*	13	1	2	1	3	1	2	6	0.36	0.42
*HIF1A*	8	0	3	0	3	0	0	6	0.36	0.36
*FGFR3*	18	2	2	2	3	1	0	11	0.33	0.34
*KRAS*	7	1	1	1	2	0	1	5	0.28	0.28
*PIK3CG*	14	1	4	1	5	1	4	7	0.22	0.23
*TP53*	24	1	16	16	18	1	9	23	0.21	0.31
*HNF1A*	13	0	5	3	4	0	0	9	0.20	0.27
*PIK3R1*	10	2	2	0	3	0	0	6	0.19	0.30
*KDM6A*	11	1	3	2	3	1	0	8	0.19	0.19
*ATM*	31	3	6	3	3	0	4	26	0.17	0.16
*MLH1*	9	1	2	1	3	0	1	8	0.16	0.17
*PRDM1*	5	0	2	0	2	0	1	3	0.16	0.16
*JAK1*	11	0	3	0	7	0	0	3	0.14	0.44

^a^ Percentage of samples with at least one pathogenic or somatic variant within the gene.

**Table 3 jpm-11-00535-t003:** Association of mutated genes with clinicopathological.

Top genes associated with female versus male (232).	**SetID**	***p*** **.Value**	**N.Marker.All**	**N.Marker.Test**	**MAC**	**m**	**Method.bin**	**MAP**
	AR	0.0002	5	5	37	19	QA	−1
	BTK	0.003	1	1	5	5	ER	0.003523
	SAMD9	0.003	14	14	66	30	QA	−1
	PAX7	0.006605	6	6	41	26	QA	−1
	KDM6A	0.010983	11	11	58	27	QA	−1
Top genes associated with young group (<50 years old) versus old (285).	5TGM7	0.002186	4	4	85	43	QA	−1
	MRE11A	0.006959	8	8	177	62	QA	−1
	NBN	0.011436	11	11	171	47	QA	−1
	VHL	0.015733	2	2	2	2	ER	0.015733
	IDH1	0.019368	7	7	16	12	ER	1.32 × 10^−10^
Top genes associated with left colorectal cancer versus right (204).	PIK3CB	0.0007	9	9	9	8	ER	8.76 × 10^−5^
	PIK3CA	0.001	12	12	86	39	QA	−1
	RNF213	0.001	61	56	709	78	MA	−1
	AURKB	0.003	5	5	101	39	QA	−1
	ERBB4	0.005	11	11	41	27	QA	−1
Top genes associated with late stage versus early stage (180).	EXT1	0.003027	7	7	73	51	MA	−1
	RNASEL	0.013275	4	4	34	29	ER.A	−1
	CDH5	0.018248	8	8	135	62	MA	−1
	BUB1B	0.021125	18	18	178	48	MA	−1
	MUTYH	0.027108	11	11	108	51	MA	−1

N.Marker.All, number of all variants within that gene; N.Marker.Test, number of variants entered the analysis (in our case, we did not exclude common variants, but we assigned them lower weight, so it will be similar as N.Marker.All); MAC, total minor allele count (MAC); m, the number of individuals with minor alleles; method.bin, a type of method to be used to compute the *p*-value; MAP, minimum possible *p*-values. The number in the bracket shows the number of effective tests (we chose to select a *p* value that is equal to 0.05/409).

**Table 4 jpm-11-00535-t004:** Candidate biomarkers from the list of 75 genes with pathogenic mutations in the large intestine.

Symbol	Entrez Gene Name	Location	Family	Entrez Gene ID for Human
APC	APC regulator of WNT signaling pathway	Nucleus	Enzyme	324
BRAF	B-Raf proto-oncogene, serine/threonine kinase	Cytoplasm	Kinase	673
CTNNB1	Catenin beta 1	Nucleus	Transcription regulator	1499
EGFR	epidermal growth factor receptor	Plasma membrane	Kinase	1956
ERBB2	Erb-b2 receptor tyrosine kinase 2	Plasma membrane	Kinase	2064
FLT1	Fms-related receptor tyrosine kinase 1	Plasma membrane	Kinase	2321
IGF1R	Insulin-like growth factor 1 receptor	Plasma membrane	Transmembrane receptor	3480
KDR	Kinase insert domain receptor	Plasma membrane	Kinase	3791
KIT	KIT proto-oncogene, receptor tyrosine kinase	Plasma membrane	Transmembrane receptor	3815
KRAS	KRAS proto-oncogene, GTPase	Cytoplasm	Enzyme	3845
MLH1	MutL homolog 1	Nucleus	Enzyme	4292
PDGFRA	Platelet-derived growth factor receptor alpha	Plasma membrane	Kinase	5156
PIK3CA	Phosphatidylinositol-4,5-bisphosphate 3-kinase catalytic subunit alpha	Cytoplasm	Kinase	5290
PTEN	Phosphatase and tensin homolog	Cytoplasm	Phosphatase	5728
SMAD4	SMAD family member 4	Nucleus	Transcription regulator	4089
TP53	Tumor protein p53	Nucleus	Transcription regulator	7157

## Data Availability

All data is available in the repository as mentioned in methods.

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
