# Peer review of "Identification of Novel Mutations in Colorectal Cancer Patients Using AmpliSeq Comprehensive Cancer Panel"

_jpm, 2021, doi:10.3390/jpm11060535_

Round 1

Reviewer 1 Report

This is an interesting original article regarding Identification of novel mutations in colorectal cancer patients using Ampliseq comprehensive cancer panel.

The topic is promising. 

I have the following comments:

The whole manuscript should be edited for english-language usage

The authors stated that "In this study, we used IonAmpliSeqTM CCP to sequence samples from archived patient samples from two hospitals in Riyadh, Saudi Arabia. The confirmation of well-known mutations point towards chromosomal instability pathway as predominant mechanism of development of CRC in this cohort. We provide comprehensive analyses of novel variants that can be useful for biomarker discovery and identification of driver genes. Discovery of biomarkers and identification of driver genes from local 86 population is critical in developing precision and personalized medicine approach towards addressing colorectal cancer".

The above-mentioned period should be strongly shortened. The authors must report only 1 sentence regarding the aim.

The period of the study must be added also in the abstract. 

The methodology of the study should be clarified (please add the correct guidelines from equator network)

Please consider the following references in both the introduction and discussion: 

  • Noninvasive Biomarkers of Colorectal Cancer: Role in Diagnosis and Personalised Treatment Perspectives. Gastroenterol Res Pract. 2018 Jun 13;2018:2397863. doi: 10.1155/2018/2397863
  • Di Nicolantonio F, Vitiello PP, Marsoni S, Siena S, Tabernero J, Trusolino L, Bernards R, Bardelli A. Precision oncology in metastatic colorectal cancer - from biology to medicine. Nat Rev Clin Oncol. 2021 Apr 16. doi: 10.1038/s41571-021-00495-z
  • Mast Cells, microRNAs and Others: The Role of Translational Research on Colorectal Cancer in the Forthcoming Era of Precision Medicine. J Clin Med. 2020 Sep 3;9(9):2852. doi: 10.3390/jcm9092852

Authors should add considerations regarding the clinical implications of their study.

Author Response

Comment: This is an interesting original article regarding Identification of novel mutations in colorectal cancer patients using Ampliseq comprehensive cancer panel. The topic is promising. 

Response: Thank you for a positive and encouraging response. We appreciate your time and effort in reviewing the manuscript.

Comment: The whole manuscript should be edited for english-language usage

Response: The whole manuscript has been checked for English language as per U.S. English language rules.

Comment: The authors stated that "In this study, we used IonAmpliSeqTM CCP to sequence samples from archived patient samples from two hospitals in Riyadh, Saudi Arabia. The confirmation of well-known mutations point towards chromosomal instability pathway as predominant mechanism of development of CRC in this cohort. We provide comprehensive analyses of novel variants that can be useful for biomarker discovery and identification of driver genes. Discovery of biomarkers and identification of driver genes from local 86 population is critical in developing precision and personalized medicine approach towards addressing colorectal cancer".

The above-mentioned period should be strongly shortened. The authors must report only 1 sentence regarding the aim.

The period of the study must be added also in the abstract. 

Response: Period has been mentioned along with the above paragraph.

Comment: The methodology of the study should be clarified (please add the correct guidelines from equator network)

Response: We checked the guidelines from equator network and confirmed that they follow the appropriate instructions.

Comment: Please consider the following references in both the introduction and discussion: 

  • Noninvasive Biomarkers of Colorectal Cancer: Role in Diagnosis and Personalised Treatment Perspectives. Gastroenterol Res Pract. 2018 Jun 13;2018:2397863. doi: 10.1155/2018/2397863
  • Di Nicolantonio F, Vitiello PP, Marsoni S, Siena S, Tabernero J, Trusolino L, Bernards R, Bardelli A. Precision oncology in metastatic colorectal cancer - from biology to medicine. Nat Rev Clin Oncol. 2021 Apr 16. doi: 10.1038/s41571-021-00495-z
  • Mast Cells, microRNAs and Others: The Role of Translational Research on Colorectal Cancer in the Forthcoming Era of Precision Medicine. J Clin Med. 2020 Sep 3;9(9):2852. doi: 3390/jcm9092852

Response: All references have been added.

Comment: Authors should add considerations regarding the clinical implications of their study.

Response: Relevant discussion has been added for clinical implications

Reviewer 2 Report

Dear Dr.  Sera Li,

In this study, entitled “ Identification of novel mutations in colorectal cancer patients using Ampliseq comprehensive cancer panel”,

In this study, Bader Almuzzaini et al we used samples to identify novel somatic variants. They described a comprehensive analysis of next generation sequencing results with a coverage of >300X. The contents are interesting. The manuscript is well written.  

Author Response

Response: Thank you very much for a positive and encouraging response. We appreciate your time and effort in reviewing the manuscript.

Reviewer 3 Report

Almuzzaini et al. identified a panel of novel mutations (466 variants) in colorectal cancer patients by using next generation sequencing and proposed some of them as biomarker candidates. I would like to congratulate the authors on a well-performed, well-designed study with a lot of supporting evidence which classify this research within the scope of personalized medicine.  The major drawback of this study is a lack of control group, but the authors have acknowledged that in study limitations. 

I have a suggestion which may depend also on the journal publication format. I had difficulties in navigating through supplementary files. Can the author, if it is acceptable by Journal policy, mark significant variations in supplementary tables where appropriate?

 Example follows:  results section (169,170 lines): "69 variants presented in at least one patient with a VAF 169 between 49-51 or 99-100, indicating that they could be germline mutations, which is also supported by the MAF (>1%)  among 45 of these variants (supplementary table S3)"- suggestion: mark by color these variants.  Additionally, may the authors name references for these criteria "VAF 169 between 49-51 or 99-100" and "MAF (>1%) ".

Statistics written as in paragraph below implicate the association analysis was duplicated (lines 215-222), which wasn’t the case, can authors rephrase a bit this part?

"By testing the association of mutated genes with clinicopathological variables, we found a significant association between ERBB2 mutation and tumor late stage (Fisher's exact t-test; p value = 0.04). Significant association between EP300 and TSHR mutations with right colon tumor (Chi-square; p = 0.02 and 0.01; respectively), and FGFR3 being more mutated in left colon  (Chi-square; p = 0.01) were also observed. For the gene-based rare variants analysis (SKAT-O), no gene was associated with clinicopathological variable at the significant threshold. However, suggestive significance was found between PIK3CB and colorectal cancer on left side (P = 0.0007), androgen receptor (AR) and female gender (P = 0.0002), TGM7 and young patient (p= 0.002), EXT1 and late stage (Table 3 A-D)".

The authors have excluded three highest mutated genes based on the presence of at least one pathogenic variant and decreasing trend in frequency of patients harboring it (Table 2). APC; RET and EGFR gene mutation. Can you specify a bit more the significance of these mutations  for the development of colorectal cancer (discussion segment)?

Can authors discus the left-sided  colon higher prevalence of mutated genes a bit more?

Figures 6 are unclear due to large data, especially 6B and 6C, is it possible to improve them, to make them more clear?

Author Response

Comment: Almuzzaini et al. identified a panel of novel mutations (466 variants) in colorectal cancer patients by using next generation sequencing and proposed some of them as biomarker candidates. I would like to congratulate the authors on a well-performed, well-designed study with a lot of supporting evidence which classify this research within the scope of personalized medicine.  The major drawback of this study is a lack of control group, but the authors have acknowledged that in study limitations. 

Response: Thank you very much for a positive and encouraging response. We appreciate your time and effort in reviewing the manuscript.

Comment: I have a suggestion which may depend also on the journal publication format. I had difficulties in navigating through supplementary files. Can the author, if it is acceptable by Journal policy, mark significant variations in supplementary tables where appropriate?

Response: The suggestion is appreciated. We have modified supplementary Table 3 accordingly.

Comment: Statistics written as in paragraph below implicate the association analysis was duplicated (lines 215-222), which wasn’t the case, can authors rephrase a bit this part?

Response: These results have been rephrased to remove ambiguity. Rephrased part reads as follows “We tested the association of mutated genes with clinicopathological variables and  found a significant association between ERBB2 mutation and tumor late stage (Fisher’s exact t-test; p value = 0.04). Mutations in EP300 and TSHR mutations were found to be significantly associated with right colon tumor (Chi-square; p = 0.02 and 0.01; respectively).Increased FGFR3 mutations was observed in left colon (Chi-square; p = 0.01)”

Comment: The authors have excluded three highest mutated genes based on the presence of at least one pathogenic variant and decreasing trend in frequency of patients harboring it (Table 2). APC; RET and EGFR gene mutation. Can you specify a bit more the significance of these mutations  for the development of colorectal cancer (discussion segment)?

Response: We have modified the discussion accordingly.

Comment: Can authors discus the left-sided colon higher prevalence of mutated genes a bit more?

Response: We have modified the discussion accordingly. Folloing text along with two references has been added. “KRAS mutations, EGFR/HER2 amplifications, and a high level of amphiregulin and epiregulin expression has been observed in left sided CRC [32][33]. Treatment strtegies widely differ according to the location of the tumor and hence understing the molecular differences in local population is pertinent.”

Comment: Figures 6 are unclear due to large data, especially 6B and 6C, is it possible to improve them, to make them more clear?

Response: Best quality figures would be available for final publication

Round 2

Reviewer 1 Report

I'm satisfied with the changes made. Congratulations